# Low Measles Vaccination Coverage and Spatial Analysis of High Measles Vaccination Dropout in Ethiopia’s Underprivileged Areas

**DOI:** 10.3390/vaccines12030328

**Published:** 2024-03-19

**Authors:** Fisseha Shiferie, Samson Gebremedhin, Gashaw Andargie, Dawit A. Tsegaye, Wondwossen A. Alemayehu, Teferi Gedif Fenta

**Affiliations:** 1Project HOPE Ethiopia Country Office, Addis Ababa P.O. Box 45, Ethiopia; gandargie@projecthope.org (G.A.); dtsegaye@projecthope.org (D.A.T.); 2School of Pharmacy, Addis Ababa University, Addis Ababa P.O. Box 1176, Ethiopia; tgedif@gmail.com; 3School of Public Health, Addis Ababa University, Addis Ababa P.O. Box 1176, Ethiopia; samsongmgs@yahoo.com; 4Project HOPE Headquarters, 1220 19th Street, NW, Suite 800, Washington, DC 20036, USA; wasefa@projecthope.org

**Keywords:** measles, vaccination, spatial distribution, dropout, hot spot analysis

## Abstract

(1) Background: Measles remains a major cause of disease and death worldwide, especially in the World Health Organization African Region. This study aimed to estimate the coverage of measles vaccinations and map the spatial distribution of measles vaccination dropout in Ethiopia; (2) Methods: A cross-sectional survey was conducted in Ethiopia’s underprivileged areas. The study included 3646 mothers/caregivers of children. ArcGIS for the spatial analysis, Global Moran’s I statistic for spatial autocorrelation, and Getis-Ord Gi* statistics for hot spot analysis were applied; (3) Results: Overall, coverages of measles-containing-vaccine first- and second-doses were 67% and 35%, respectively. Developing regions had the lowest coverages of measles-containing-vaccine first- and second-doses, 46.4% and 21.2%, respectively. On average, the measles vaccination dropout estimate was 48.3%. Refugees had the highest measles vaccination dropout estimate (56.4%). The hot spot analysis detected the highest burden of measles vaccination dropout mainly in the northeastern parts of Ethiopia, such as the Afar Region’s zones 1 and 5, the Amhara Region’s North Gondar Zone, and peripheral areas in the Benishangul Gumuz Region’s Assosa Zone; (4) Conclusions: The overall measles vaccination coverages were relatively low, and measles vaccination dropout estimates were high. Measles vaccination dropout hot spot areas were detected in the northeastern parts of Ethiopia.

## 1. Introduction

Measles remains a major cause of disease and death worldwide, especially in the World Health Organization (WHO) African Region (AFR) [1]. During 2017–2021, regional coverage of the measles-containing-vaccine first-dose (MCV1) remained stable at 68–70% but below the level necessary to achieve high population immunity against measles (≥95%). The low coverage of MCV1 in populous countries like the Democratic Republic of the Congo, Ethiopia, and Nigeria has largely contributed to AFR’s low coverage [2,3]. Despite the existence of an effective vaccine, regions have varying success in measles control [4].

Currently, infants are receiving 11 different antigens from the Expanded Program for Immunization of Ethiopia through a combination of outreach, static, and mobile activities; services are primarily offered in health centers and health posts. Large-scale, periodic campaigns are also being used to administer the vaccines against meningitis, polio, and measles [5]. Ethiopia introduced the measles-containing-vaccine second-dose (MCV2) program in 2018 and launched it into the routine immunization program in 2019 to boost immunity and halt repeated measles outbreaks. MCV1 is given at nine months after birth for all children, and MCV2 is given at fifteen months [6,7].

Based on the Ethiopian Mini Demographic and Health Survey (EMDHS) 2019 report, the coverage of MCV1 was 59%, and the lowest coverage was in the Afar Region (28.5%) [8]. There was a persistently high incidence of measles in the Oromia Region of Ethiopia between 2011 and 2018, where 25,000 measles cases were reported [9]. A study that used the EMDHS 2019 data showed that the MCV2 coverage was 12.36% [10]. A 2022 study conducted in urban areas of the North Showa Zone of central Ethiopia estimated an MCV2 coverage of 42.5%. The primary reason for not receiving the MCV2, according to the study, was a lack of awareness of the need to return for a second dose of the measles vaccination [11]. On the other hand, a joint report of WHO and UNICEF indicated that the coverage of MCV2 in Ethiopia was 46% and 48% in 2021 and 2022, respectively [12].

Per the WHO’s recommendation, the main indicators of immunization to estimate dropout include discrepancies between the first and third doses of the Diphtheria–Tetanus–Pertussis (DTP1 to DTP3), Bacillus Calmette–Guérin (BCG) vaccine to MCV1, and MCV1 to MCV2 [13]. The continuation of the childhood immunization program is evaluated using vaccination dropout rates. A dropout rate of more than 10%, according to the WHO, is undesirable and suggests that a health facility has utilization constraints. Conversely, a low dropout rate suggests high service quality due to good utilization [14,15,16]. Vaccination dropout using DTP1 to DTP3 and BCG to MCV1 using the same dataset had already been published. The research revealed that caregivers who worked outside the home, belonged to the lowest socioeconomic classes, did not receive follow-up care from a skilled birth attendant, did not receive postnatal care, were older than 45 years old, and lacked gender empowerment were more likely to experience vaccination dropout [17].

To eliminate measles by 2020, AFR countries and partners planned to achieve  ≥95% with two doses of MCV coverage and improve the quality of supplemental immunization activities [8,18]. However, these goals were not met and, in 2021, coverages of MCV1 and MCV2 remained <95%. To reach the updated 2030 regional measles elimination target in at least 80% of African countries, it is imperative to administer both doses of the measles vaccine to every child and enhance surveillance [8]. Estimating the coverage of measles vaccination and understanding the geographical distribution of measles vaccination dropout is crucial to setting up appropriate interventions. This study was carried out to estimate the coverage and dropout rate of measles vaccination and map the spatial distribution of high- and low-risk areas for vaccination dropout in remote and underserved settings of Ethiopia.

## 2. Materials and Methods

### 2.1. Study Design and Settings

The data for this study were taken from a large cross-sectional evaluation survey that was conducted from May to July 2022. Areas with zero-dose and under-immunized children were first identified through a qualitative situational analysis using secondary data sources prior to the quantitative survey.

With the understanding that zero-dose and under-immunized children in low-income countries are located in underserved areas [19], the study targeted eight partly overlapping settings. In Ethiopia, developing regions are those that are governed by relatively young administrations and need more assistance to raise their level of development to that of the country’s other developed regions. In addition to having less developed infrastructure, these regions’ development in many other areas is progressing slowly, and border-related conflicts impede development even more. Newly established regions are those that were formed between June 2020 and November 2021. These include Sidama and Southwest Ethiopia and they were split off from Southern Nations, Nationalities, and Peoples (SNNP) (see Table 1 and Figure 1) [17,20].

### 2.2. Study Participants

The target population for this study was children under five who lived in the identified eight partly overlapping settings. Children in the age range of 1 and 3 years were included in the study.

### 2.3. Sample Size Determination

The WHO 2018 immunization coverage cluster survey manual was used as a reference for the sampling design of the study [21]. The appropriate sample size for each target population was calculated using Cochran’s Single Proportion Sample Size Formula [22] considering a 95% confidence level, 4% margin of error, 16% prevalence of zero-dose children [6], and 10% compensation for possible non-response. The number of children needed for a prevalence study was determined using the sample size formula below: $n=\frac{Z2p\left(1-p\right)}{d2}$, where *n* is the total sample size required, *Z* is the statistic corresponding to a 95% confidence level, which is 1.96, *p* represents the prevalence of children receiving zero doses in earlier studies conducted in Ethiopia, and *d* is precision (corresponding to effect size) [23]. Therefore, a sample size of 360 was required for every target population domain.

Based on data from the Ethiopian DHS 2016 and EMDHS 2019, there were, on average, 12 children aged 12–35 months in each enumeration area [6]. Under the assumption that every child in the EA would be qualified for study participation, a minimum of 30 EAs were needed for every population domain in order to recruit 360 children in each target population. Forty EAs were chosen at random from urban slums, and 480 children were selected [17,20].

With 360 samples per population domain, the study originally planned to include 4080 children from 340 EAs, but because of security concerns in some study districts, only 3646 children between the ages of 12 and 35 months were included in the actual survey. The overall sample size was sufficient to allow for the analysis of subgroups according to age, sex, and other pertinent background variables [17,20].

### 2.4. Sampling Procedure

To choose children between the ages of 12 and 35 months, a two-step procedure was employed in a cluster sampling approach. First, a random selection of EAs was made from all of the EAs that were available in each target population domain. The sampling frame was the EAs as defined by the Ethiopian Central Statistical Agency [6]. EA maps were created by skilled cartographers after hot spot urban slums in Addis Ababa, Adama, Bahir Dar, Hawassa, Harar, and Dire Dawa cities were identified and demarcated. Villages or clusters were considered as EAs for the IDP and refugee camps. After all of the eligible children in each EA were listed, 12 children were selected using a smartphone-based random number generator for inclusion in the study [17,20].

### 2.5. Data Collection Procedures and Data Quality Assurance

Data were gathered using pretested tools prepared in the languages of Afar, Somali, Amharic, Afan Oromo, and Sidama. The CommCare digital app version 2.52.1, an application system that is open-source, user-friendly, and compatible with popular data analytics and visualization software, was used by 48 seasoned enumerators and 24 supervisors to collect survey data. The app assisted in collecting data on individual children and households in order to ensure high-quality data collection, cleaning, and real-time monitoring [17,20].

Enumerators and supervisors who participated in the survey had at least a diploma, a great deal of experience with comparable national surveys, and knowledge of the CommCare digital app. Before being deployed, they attended a five-day training program that was led by a structured training manual. Mock interviews, field exercises, an introduction to using the CommCare digital app, an explanation of basic ethical research practices involving human subjects, the sampling approach, fundamental data collection principles, and a line-by-line discussion of the questionnaire were all covered in the training. The questionnaire was divided into ten sections excluding the information sheet and consent form section. These include basic information about the interview, sociodemographic information, household characteristics, access to health services, maternal health service utilization, knowledge and attitudes on vaccination, child immunization history, barriers and enablers to vaccination, service integration, and gender empowerment sections [17,20].

Enumerators could collect data from up to six mothers/caregivers per person per day. Supervisors conducted follow-up interviews with one-third of the study participants to confirm the quality of the data. In addition, the research team closely monitored the uploaded data throughout the survey implementation period [17,20].

### 2.6. Ascertainment of Childhood Vaccination

In accordance with the WHO recommendations, mothers’/caregivers’ reports, immunization cards, and facility records were utilized to ascertain children’s vaccination status. When presented by the mother or caregiver, the vaccination card was used to determine the child’s vaccination status. If this card was not presented, the mother or caregiver’s self-report was used to determine the child’s immunization status. Prior research has confirmed this method’s dependability for determining childhood immunization in resource-constrained environments with insufficient childhood immunization documentation [24].

### 2.7. Data Management and Statistical Analysis

Every day, data were gathered via the CommCare digital app [25] and kept on a local server. The CommCare digital app has built-in features to support the global positions system’s data collection at the field level. The descriptive and summary statistics, such as cross-tabulations and frequency tables, were generated using STATA version 17.0 (Stata Corp. Statistical Software) [26]. ArcGIS version 10.8 was used for the spatial analysis. The data were weighted to make the survey representative. In order to balance weighted and unweighted sample sizes, linearization of post-stratification weights was made.

### 2.8. Measles Vaccination Coverage

Coverage was calculated as the proportion of children aged 12–35 months who received either MCV1 or MCV2. Overall MCV1 and MCV2 vaccination coverage estimates, as well as coverage estimates separately for each population domain, were computed.

### 2.9. Measles Vaccination Dropout

The percentage of children who received the first vaccination but did not receive the second vaccine is known as the vaccination dropout estimate. For this study, the measles vaccination dropout estimate was defined as the percentage of children who did not receive MCV2 among those who received MCV1. For the spatial analysis, measles vaccination dropout cases by the zonal administrative unit were mapped.

### 2.10. Spatial Autocorrelation

The distribution pattern of measles vaccination dropout cases across the study area was examined using spatial autocorrelation to determine whether dropout was dispersed, clustered, or randomly distributed. The Global Moran’s I spatial statistics was used to measure spatial autocorrelation, whereby a value approaching +1 would indicate spatially clustered measles vaccination dropout cases and a value approaching −1 would suggest a dispersed spatial distribution of measles vaccination dropout cases. A Moran’s I value of 0 would indicate a random geographic distribution of measles vaccination dropout cases. A statistically significant Moran’s I (*p*-value < 0.05) would strongly suggest the presence of spatial autocorrelation and would likely lead to the rejection of the null hypothesis (measles vaccination dropout is randomly distributed).

### 2.11. Hot Spot Analysis (Getis-Ord Gi* Statistics)

A hot spot analysis using Getis-Ord Gi* statistics was applied to show the spatial variability of measles vaccination dropout cases. A z-score with a 95% confidence interval and a *p*-value < 0.05 confirmed the statistical significance of clustering. Statistical output with a high Gi* would signify measles vaccination dropout hot spots (i.e., more significant numbers of dropout children), while a low Gi* would suggest measles dropout cold spots (i.e., lower numbers of dropout children).

### 2.12. Ethical Approval

The research was implemented in compliance with national and international ethical principles. The research protocol was reviewed and approved by the institutional review board of the Ethiopian Public Health Institute (416/2021). Data were collected after obtaining written informed consent from the mothers/caregivers. All zero-dose children were referred to the nearest health facility using a referral form to maximize beneficence.

## 3. Results

### 3.1. Sociodemographic Characteristics

The data for this study were taken from a large cross-sectional evaluation survey, and the key findings of the original survey have already been published. With a response rate of 97.7%, 3646 mothers/caregivers of children between the ages of 12–35 months participated in the study. Among the respondents, the majority (59.2%) had no formal education, and over half (54%) were between the ages of 25–34 years. More than 81% came from rural areas, with 17% coming from the Afar region. When the survey was conducted, 57% of the respondents were unemployed, and nearly 91% of the respondents were married or living together (Table 2) [17,20].

### 3.2. Coverage of Measles in Remote and Underserved Settings of Ethiopia

Coverages of MCV1 and MCV2 varied across the different target populations. The overall coverage of MCV1 was 67% and that of MCV2 was 35%. Urban slums had the highest MCV1 and MCV2 coverages, 86.8% and 54.1%, respectively. On the other hand, the lowest MCV1 and MCV2 coverages were in developing regions, 46.4% and 21.2%, respectively. Figure 2 below shows the numbers and proportions of children who had received the measles vaccines according to vaccination cards, medical records, or mothers’/caregivers’ self-reports.

### 3.3. Measles Vaccination Dropout Estimates in Remote and Underserved Settings of Ethiopia

The different target populations had varying measles vaccination dropout estimates. Overall, the measles vaccine dropout estimate was 48.3%. While the highest measles dropout estimate was in refugees (56.4%), urban slums had the lowest measles dropout estimate (37.7%) (Table 3).

### 3.4. Spatial Autocorrelation Analysis

The spatial autocorrelation analysis revealed that the spatial distribution of measles vaccination dropout cases in Ethiopia was non-random (Global Moran’s I = 0.992195, *p* < 0.001). The result showed that the observed Global Moran’s I value was greater than the expected index (−0.000046) and was statistically significant. This clustered pattern has a less than 1% probability of being the result of random chance, according to the z-score of 205.881092 (Figure 3).

### 3.5. Hot Spot Analysis of Measles Vaccination Dropout Cases

The geographical distributions of measles vaccination dropout cases by zonal administrations were identified using the Getis-Ord GI* hot spot statistical analysis technique. The dark red color indicates significant (*p* < 0.001) clusters of high measles vaccination dropout (risk areas), the orange color shows zones with significant (*p* < 0.001) clusters of medium-level measles dropout cases, and the gray color indicates significant (*p* < 0.001) clusters of low measles vaccination dropout (non-risky areas). As shown in Figure 4 below, the highest burden of measles vaccination dropout cases (hot spot areas) were detected in zones 1 and 5 of the Afar Region, the North Gondar Zone of the Amhara Region, peripheral areas in the Assosa Zone of the Benishangul Gumuz Region, the Agnewak Zone of the Gambella Region, certain areas of the Harari Region, the Siti Zone of the Somali Region, the West Hararge Zone of the Oromia Region, the South Omo Zone of the SNNP Region, and in certain areas of the Addis Ababa City Administration. A relatively higher burden of measles vaccination dropout cases was also found in the West Gojjam and Waghmra zones of the Amhara Region, the Nuwer Zone of the Gambella Region, the East Wellega, East Shewa, Borena, and East Hararge zones of the Oromia Region, peripheral areas of the Addis Ababa City Administration, the Fafan and Liban zones of the Somali Region, and rural areas of Dire Dawa City. On the other hand, most parts of the Oromia, Somali, SNNP, and Amhara regions and the Southwest Ethiopia and Sidama regions were identified as cold spot areas for measles vaccination dropout cases (non-risky areas) (Figure 4).

## 4. Discussion

A total of 3646 mothers/caregivers with children aged 12–35 months from 340 clusters were included in the study. The coverage of childhood MCV1 and MCV2 were 67% and 35%, respectively, which were higher than the estimates from the EMDHS 2019 data of 59% for MCV1 [8,27] and 12% for MCV2 [10]. However, the MCV2 coverage result from our study was much lower than the finding presented in the 2022 joint report of the WHO and UNICEF (48%) [12]. Our study’s inclusion of underprivileged areas could possibly explain this discrepancy between MCV2 coverage estimates.

The coverage estimates for the measles vaccine varied between the different population domains. Among these target populations, urban settings had the highest MCV1 and MCV2 coverage estimates. Compared to previous studies in urban areas, the MCV2 coverage result showed a modest improvement. For instance, in underserved urban areas, the estimated coverage of MCV2 in the present study was 54.1%, which was higher than the coverage in urban areas of the North Shoa Zone in central Ethiopia (42.5%) [11]. All the childhood MCV1 and MCV2 coverages of the present study were still far below the recommended herd immunity threshold (95%) necessary to achieve high population immunity against measles. This indicates that Ethiopia and the AFR are still at high risk for measles outbreaks [28].

Overall, the measles vaccination dropout estimate was 48.3%. This was higher than the vaccination dropout estimates from the same study looking at DTP1 to DTP3 (44%) and BCG to MCV1 (21.5%) [17]. Similar to the vaccination coverage results, the measles vaccination dropout estimates varied across the different target populations. Higher measles vaccination dropout estimates were found in refugees (56.4%), pastoralist populations (54.3%), developing regions (54.4%), and newly established regions (53.5%), whereas the lowest estimate was in urban slums (37.7%). These findings are in line with a previous study that reported higher dropout estimates in pastoralist regions [29]. The present study’s findings are also consistent with a study by Shiferie et al. which utilized the same dataset and found the highest vaccination dropout estimate was in newly established regions (73%) based on BCG and MCV2, whereas the lowest vaccination dropout estimates were in urban slums [17]. The delayed introduction of MCV2 into Ethiopia’s routine immunization program in 2020 might have contributed to the increased MCV1 to MCV2 dropout estimate. Moreover, a lack of awareness of the need to return for MCV2 may have also led to a higher dropout estimate [11]. All the measles vaccination dropout estimates of the present study were higher compared to the WHO’s 10% maximum acceptable dropout threshold. This could be a sign of the immunization programs’ inadequate performance in Ethiopia’s underprivileged areas [14,16].

Donors and policymakers are increasingly interested in mapping the spatial heterogeneity of childhood vaccination to identify existing gaps and intervene accordingly. The global spatial autocorrelation analysis of our data showed a clustering pattern of childhood measles vaccination dropout cases across the study areas (Global Moran’s I = 0.992195, *p* < 0.001). This indicated that measles vaccination dropout cases were aggregated in specific areas. The hot spot analysis showed that the highest burden of measles vaccination dropout cases (hot spot areas) were detected in zones 1 and 5 of the Afar Region, the North Gondar Zone of the Amhara Region, peripheral areas in the Assosa Zone of the Benishangul Gumuz Region, the Agnewak Zone of the Gambella Region, certain areas of the Harari Region, the Siti Zone of the Somali Region, the West Hararge Zone of the Oromia Region, the South Omo Zone of the SNNP Region, and certain areas of the Addis Ababa City Administration.

The findings of the hot spot analysis are consistent with previous surveys and studies conducted in Ethiopia. A study on the spatial distribution of immunization defaulting in Ethiopia showed that most defaulters were in the eastern and northeastern parts of the country [30]. The previous literature also indicated that areas with low immunization were spatially clustered in the northeastern parts of the country [31,32,33,34]. Moreover, two separate studies by Tesfa et al. also showed the presence of low vaccination clusters in zones 1, 4, and 5 of the Afar Region and the Liben Zone of the Somali Region [27,35]. The high burden of measles vaccination dropout cases in northeastern parts of Ethiopia was attributed to the nomadic and pastoralist populations that move from place to place seasonally, negatively inhibiting access to health care services, including immunization [36,37,38,39]. Low utilization of health information in these parts of Ethiopia might also contribute to the high burden of measles vaccination dropout cases [40]. On the other hand, most parts of Addis Ababa, Oromia, Somali, SNNP, and Amhara regions and Southwest Ethiopia and Sidama regions were identified as cold spot areas for measles vaccination dropout cases (non-risky areas).

This study was strong in a number of ways. The research team collected data in areas where there were ongoing conflicts. Furthermore, data collected from a variety of sources, such as vaccination cards, medical records, and mother/caregiver recall, were used to determine childhood vaccination dropout. The results were validated, and the data quality was strengthened by this triangulation. The collection of high-quality data was also facilitated by the utilization of digital tools and the employment of skilled data collectors. Hot spot areas for measles vaccination dropout cases were identified using advanced geospatial analysis techniques.

However, the study also had limitations, including vulnerability to biases that are closely related to the cross-sectional study design, such as recall bias and nonresponse bias. Mothers or caregivers who lacked vaccination cards may have forgotten their child’s immunization status, which could have led to misclassification.

## 5. Conclusions and Recommendations

The overall MCV1 and MCV2 coverage estimates were low, measles vaccination dropout was high, and there were geographical variations in both parameters across Ethiopia. The hot spot analysis showed that the highest burdens of measles vaccination dropout cases (hot spot areas) were detected mainly in the northeastern parts of Ethiopia.

Due to its recent introduction, the lack of awareness around MCV2 needs to be addressed in Ethiopia, especially in the northeastern part of the country where there were the highest estimates of measles vaccination dropout.

This study aimed to contribute to improving Ethiopia’s measles vaccination coverage by providing evidence for targeted interventions. Moreover, this research could help fulfill the 2030 immunization agenda, which aims to provide everyone in the world with equitable and effective vaccines to tackle vaccine-preventable diseases in order to improve health and wellbeing.

## Figures and Tables

**Figure 1 vaccines-12-00328-f001:**
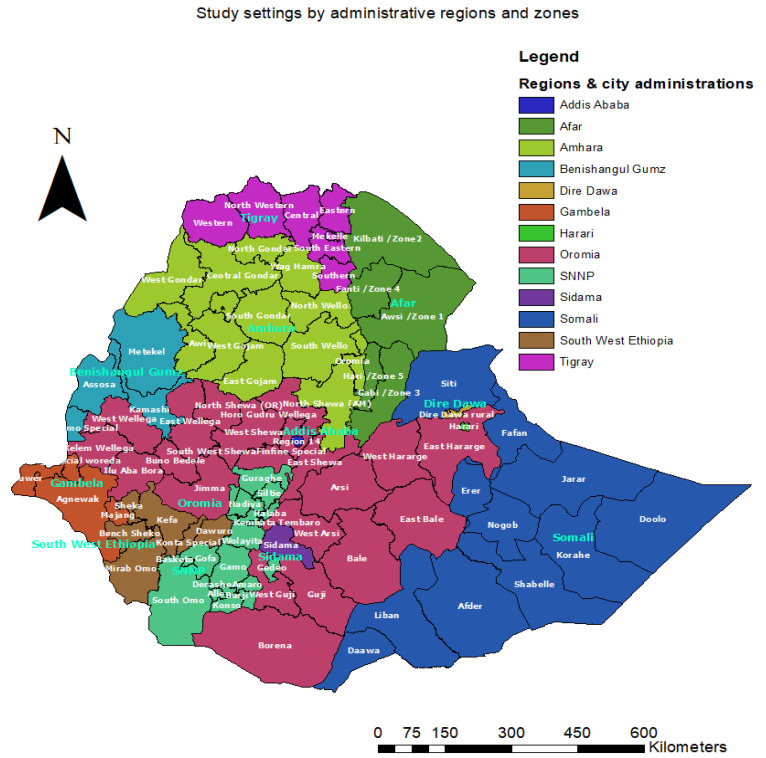
Study settings by administrative regions and zones.

**Figure 2 vaccines-12-00328-f002:**
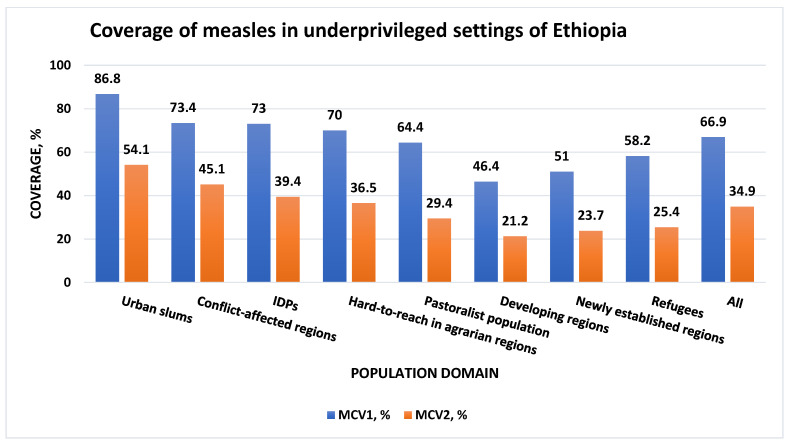
MCV1 and MCV2 vaccination coverages in underserved settings of Ethiopia.

**Figure 3 vaccines-12-00328-f003:**
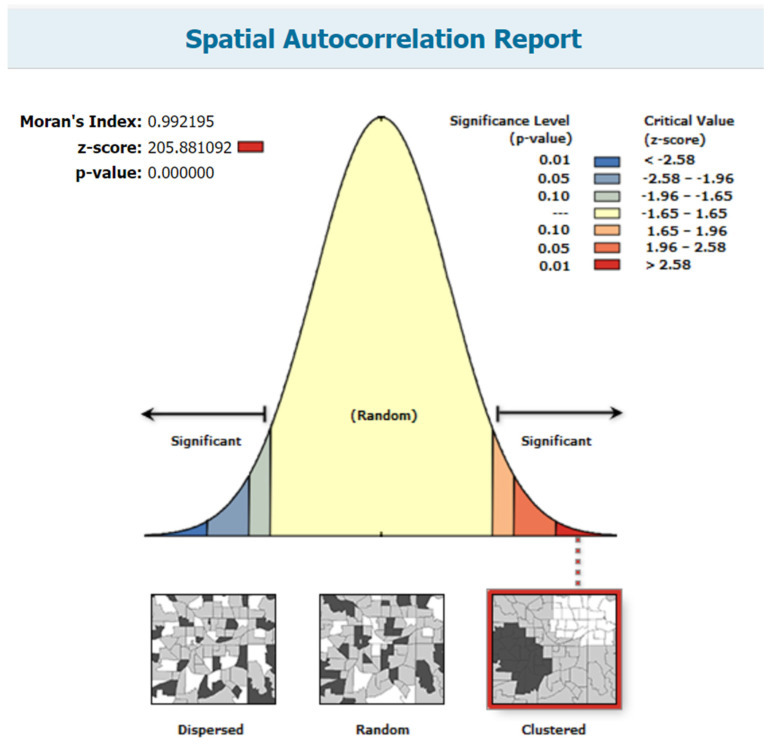
Spatial autocorrelation analysis of measles vaccination dropout cases in Ethiopia.

**Figure 4 vaccines-12-00328-f004:**
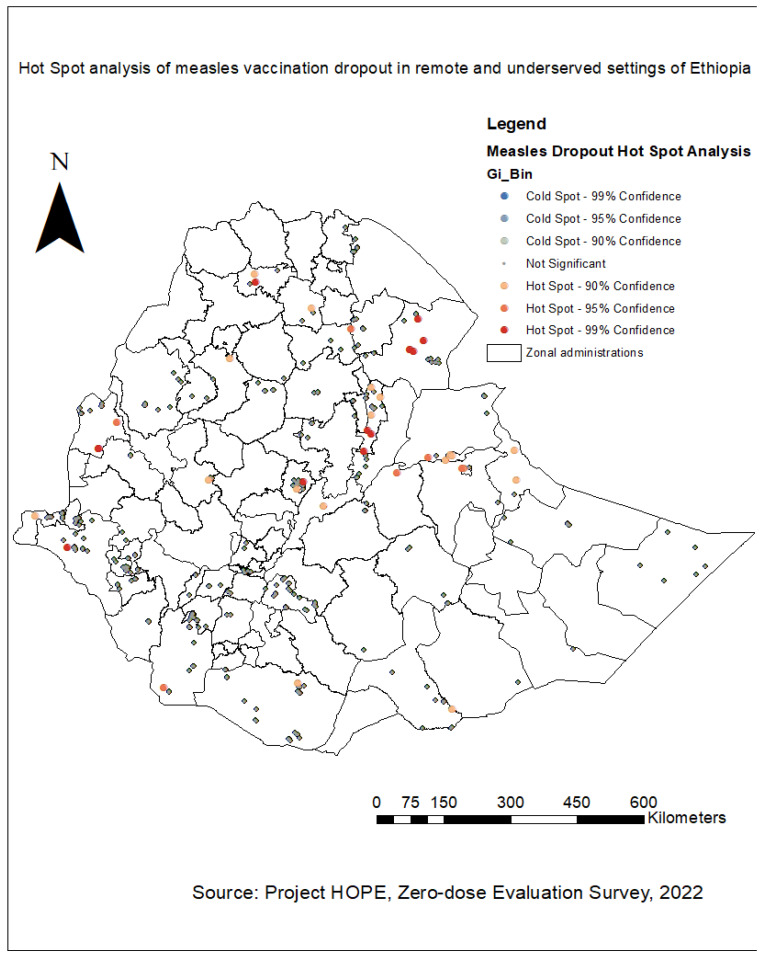
Hot spot analysis of measles vaccination dropout cases in Ethiopia [20].

**Table 1 vaccines-12-00328-t001:** Target population domains and their descriptions.

	Population Domain	Description
1	Pastoralist regions and populations	Afar and Somali regions and specific pastoralist or semi-pastoralist settings in Oromia, SNNP, Southwest Ethiopia, and Gambella regions
2	Developing regions	Afar, Somali, Gambella, and Benishangul Gumuz regions
3	Newly established regions	Sidama and Southwest Ethiopia regions
4	Conflict-affected areas	Selected settings in Afar, Amhara, Oromia, and Benishangul Gumuz regions
5	Underserved urban slums	Urban slums in six selected cities (Addis Ababa, Bahirdar, Hawassa, Dire Dawa, Harar, and Adama) and rural areas under Dire Dawa City Administration and Harari Region
6	Hard-to-reach areas in major regions	Selected remote districts in Amhara, Oromia, and SNNP regions
7	Internally displaced populations (IDPs)	Selected IDP centers in Afar, Amhara, Oromia, and Benishangul Gumuz regions
8	Refugees	Refugees from selected camps in Somali, Afar, and Gambella regions

**Table 2 vaccines-12-00328-t002:** Sociodemographic characteristics of mothers/caregivers and their children in underserved settings of Ethiopia, 2022.

Characteristics	Frequency	Percent
Child’s sex		
Male	1985	54.4
Female	1661	45.6
Child’s age (months)		
12–23	1849	50.7
24–35	1797	49.3
Mother’s/Caregiver’s age (years)		
15–24	875	24.0
25–34	1969	54.0
35–44	572	15.7
45 or above	104	2.9
Do not know	126	3.5
Mother’s/Caregiver’s educational status		
No formal education or preschool	2158	59.2
Primary education	788	21.6
Secondary education	616	16.9
Tertiary education	84	2.3
Marital status		
Not ever married	43	1.2
Married/Living together	3312	90.8
Separated	83	2.3
Divorced	110	3.0
Widowed	98	2.7
Place of residence		
Urban	677	18.6
Rural	2969	81.4
Caregiver’s employment status		
Unemployed	2098	57.6
Employed	1548	42.4
Region *		
Afar	636	17.4
Amhara	372	10.2
Oromia	431	11.8
Somali	480	13.2
Benishangul Gumuz	216	5.9
Southern Nations, Nationalities, and Peoples	300	8.2
Sidama	239	6.6
Southwest Ethiopia	181	5.0
Gambella	479	13.1
Harari	60	1.6
Addis Ababa	192	5.3
Dire Dawa	60	1.6
Household size		
2–5	2044	56.0
6 or above	1602	44.0

* Unweighted sample size.

**Table 3 vaccines-12-00328-t003:** Measles vaccination dropout estimates across the target populations in Ethiopia, 2022.

Target Population Domain	MCV1 to MCV2 Dropout Estimate * *n* (%)
Urban slums	145 (37.7)
Conflict-affected regions	78 (38.5)
IDPs	89 (46.2)
Hard-to-reach in agrarian regions	264 (47.9)
Pastoralist population	469 (54.3)
Developing regions	348 (54.4)
Newly established regions	114 (53.5)
Refugees	102 (56.4)
All ^†^	1158 (48.3)

^†^ Target population groups add up to more than the total because certain populations contributed to multiple domains. * Table is only looking at participants identified as dropouts for measles from the larger study population—it does NOT account for all 3646 participants.

## Data Availability

The data presented in this study are available on request from the corresponding author.

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
