# Peer review of "Low Measles Vaccination Coverage and Spatial Analysis of High Measles Vaccination Dropout in Ethiopia’s Underprivileged Areas"

_vaccines, 2024, doi:10.3390/vaccines12030328_

Round 1

Reviewer 1 Report

Comments and Suggestions for Authors

In this paper, the authors leveraged a subset of data collected from a large cross-sectional survey on vaccination in Ethiopia to evaluate the vaccination coverage of measles and the dropout rates and to identify hot spots of high dropout in underserved regions of Ethiopia. They found that measles vaccination coverage was low overall, with high dropout rates in Ethiopia. Hot spot areas for measles vaccination dropouts were mainly identified in northeastern Ethiopia. These results would help design tailored vaccination strategies for specific areas in the future to improve the vaccination coverage in underserved areas of Ethiopia. However, part of the methodology and presentations of the results could be improved to make the major results clearer to the readers. Therefore, I would suggest the authors address the following comments before this paper can be considered for publication.

Specific comments

Introduction

1.     Line 34: spell ‘AFR’ out as it first appears in the text.

2.     Line 68: is it MCV2 coverage or MCV1 coverage that was planned to achieve > 95%?

Methods

3.     Table 1: A map is needed here to facilitate the readers to better understand the geographical locations of these varying regions. Also, it would be also helpful to provide the definitions of these population domains, such as developing regions, and newly established regions.

4.     Line 128: what variables were included in the survey? Do you have a sample questionnaire as a supplementary material?

Results

5.     Figure 1: The statistics of Global Moran's I described in the text were sufficient; a screenshot of the output from the software is usually not necessary. Instead, before performing the Global Moran's I, it would be better to first present the geographical distribution of the dropout rates on a map.

6.     Line 232: You said the hot spots were identified at the zonal administration level, but I didn't see that you aggregated your point data to the regional levels. What does each point in Figure 3 represent? Some zones have only one point, but others have multiple points. Did only one point can represent the whole zone? These results are not clear to me.

7.     Lines 238-242: How did these regions correspond to your defined eight population domains?

8.     Figure 3: this figure can be improved by separating the labels and the points. You could consider using abbreviations on your map. The busy text on the map makes it hard to see the overall spatial pattern.

Author Response

Dear Reviewer,

We thank you so much for your time and expertise. Your comments have been invaluable in improving the content of our manuscript. We have revised our manuscript in track changes based on the comments.  

Kindly please see the attachment for the point-by-point responses to your comments. Responses/changes are highlighted or indicated with line numbers.

Kind regards,

Fisseha Shiferie

Reviewer 2 Report

Comments and Suggestions for Authors

This work focuses on low measles vaccination coverage and the spatial analysis of high measles dropout in Ethiopia's underprivileged areas. The authors used a survey to gather their data and spatial analysis techniques were used to observe the spatial distribution of vaccination in Ethiopian communities.

Major comment

Is there no measles vaccination data in Ethiopia? If there is, why can't authors validate their survey with real data? I am of the opinion that a survey is an opinion and sometimes doesn't capture the real event, especially when you are dealing with less educated communities.

Minor comment 

Table 2 should move to appendix

Abbreviations in the abstract should be written in full since they are appearing for the first time in the text. 

Author Response

Dear Reviewer,

We thank you so much for your time and expertise. Your comments have been invaluable in improving the content of our manuscript. We have revised our manuscript in track changes based on the comments.

Kindly please see our point-by-point responses in the attachment. Changes are highlighted or indicated with line numbers.

Kind regards,

Fisseha Shiferie

Round 2

Reviewer 1 Report

Comments and Suggestions for Authors

See my additional comments:

4. Line 128: what variables were included in the survey? Do you have a sample questionnaire as a supplementary material?

Answer: Some of the variables of our study include wealth index, marital status, child age, respondent age, time to walk to the health facility (one-way), maternal educational status, paternal educational status, caregiver’s employment status, antenatal care visits, postnatal care (PNC) services, place of residence, number of under-five children, skilled birth attendance (SBA), availability of health facility in the kebele (a small administrative unit), gender empowerment, child sex, and sex of household head. Yes, we have a questionnaire and we can share it if needed. 

Comment: Then add a brief description of your questionnaire in the main text.

5. Figure 1: The statistics of Global Moran's I described in the text were sufficient; a screenshot of the output from the software is usually not necessary. Instead, before performing the Global Moran's I, it would be better to first present the geographical distribution of the dropout rates on a map.

Answer: We agree with the reviewer’s feedback but we included the screenshot of the output to enable the map to speak for itself and enhance visualization effect. Published spatial analysis articles we have read/referred so far also follow the same result presentation and we followed the same procedures. Therefore, below are the procedures we followed for our manuscript. The first thing that should be done is examining the distribution pattern of measles vaccination dropout across the study areas to see whether dropout was dispersed, clustered, or randomly distributed using spatial autocorrelation. Once the pattern is examined, other spatial analyses can follow. This is the procedure many published articles follow. 

Comment: Ok, you can keep the output from ArcGIS. But, based on the procedure that you followed, what is the distribution pattern that you tested on? It would be better that you can first show the distribution of dropout rates at the zonal level in a map before you performed the spatial autocorrelation test. That map can be panel A, and spatial autocorrelation result is panel B in Figure 3.

6. Line 232: You said the hot spots were identified at the zonal administration level, but I didn't see that you aggregated your point data to the regional levels. What does each point in Figure 3 represent? Some zones have only one point, but others have multiple points. Did only one point can represent the whole zone? These results are not clear to me.

Answer: Each point in Figure 3 represents measles vaccination dropout. Actually, the number of points is more than what is displayed on the map. As we zoom in the map, the number of points keep increasing. A certain zone that has a smaller number of measles dropout will only have a few points. One point does not necessarily represent the whole zone. As described earlier, a zone that has more measles dropouts will have many points and if it has less dropouts, it will be represented by few points.  

Comment: You said that the measles dropouts were represented by the number of points in each zone. Does that mean these points were just randomly generated by the software for each zone but didn't have genuine coordinates? If so, I don't think it's appropriate to use the Getis-Ord to identify hotspots in this case. Rather, you can calculate a dropout rate for each zone and use the zonal administrative area as the geographic unit to perform the Getis-Ord test.

Author Response

Point-by-point responses to comments of reviewer 1

Dear Reviewer,

We thank you so much again for your time and expertise. We have revised our manuscript in track changes based on your comments. Kindly please see below our point-by-point responses in italics. We have included the spatial distribution of Measles vaccination dropout in our manuscript as Figure 3a which can also be found on page 3 in the attached cover letter. 

  1. Line 128: what variables were included in the survey? Do you have a sample questionnaire as a supplementary material?

Answer:

Some of the variables of our study include wealth index, marital status, child age, respondent age, time to walk to the health facility (one-way), maternal educational status, paternal educational status, caregiver’s employment status, antenatal care visits, postnatal care (PNC) services, place of residence, number of under-five children, skilled birth attendance (SBA), availability of health facility in the kebele (a small administrative unit), gender empowerment, child sex, and sex of household head.

Yes, we have a questionnaire and we can share it if needed.

Comment: Then add a brief description of your questionnaire in the main text.

Answer:

The questionnaire was divided into ten sections excluding the information sheet and the consent form section. These include basic information about the interview, socio-demographic information, household characteristics, access to health services, maternal health service utilization, knowledge and attitudes on vaccination, child immunization history, barriers and enables to vaccination, service integration and gender empowerment sections (lines 152- 157).

Results

  1. Figure 1: The statistics of Global Moran's I described in the text were sufficient; a screenshot of the output from the software is usually not necessary. Instead, before performing the Global Moran's I, it would be better to first present the geographical distribution of the dropout rates on a map.

Answer:

We agree with the reviewer’s feedback but we included the screenshot of the output to enable the map to speak for itself and enhance visualization effect. Published spatial analysis articles we have read/referred so far also follow the same result presentation and we followed the same procedures. Therefore, below are the procedures we followed for our manuscript.

The first thing that should be done is examining the distribution pattern of measles vaccination dropout across the study areas to see whether dropout was dispersed, clustered, or randomly distributed using spatial autocorrelation. Once the pattern is examined, other spatial analyses can follow. This is the procedure many published articles follow.

Comment: Ok, you can keep the output from ArcGIS. But, based on the procedure that you followed, what is the distribution pattern that you tested on? It would be better that you can first show the distribution of dropout rates at the zonal level in a map before you performed the spatial autocorrelation test. That map can be panel A, and spatial autocorrelation result is panel B in Figure 3.

Answer:

We have done the spatial distribution of measles vaccination dropout. Figure 3a shows the spatial distribution of measles vaccination dropout and Figure 3b depicts the spatial autocorrelation.

  1. Line 232: You said the hot spots were identified at the zonal administration level, but I didn't see that you aggregated your point data to the regional levels. What does each point in Figure 3 represent? Some zones have only one point, but others have multiple points. Did only one point can represent the whole zone? These results are not clear to me.

Answer:

Each point in Figure 3 represents measles vaccination dropout. Actually, the number of points is more than what is displayed on the map. As we zoom in the map, the number of points keep increasing.

A certain zone that has a smaller number of measles dropout will only have a few points. One point does not necessarily represent the whole zone. As described earlier, a zone that has more measles dropouts will have many points and if it has less dropouts, it will be represented by few points.

Comment: You said that the measles dropouts were represented by the number of points in each zone. Does that mean these points were just randomly generated by the software for each zone but didn't have genuine coordinates? If so, I don't think it's appropriate to use the Getis-Ord to identify hotspots in this case. Rather, you can calculate a dropout rate for each zone and use the zonal administrative area as the geographic unit to perform the Getis-Ord test.

Answer:

Every point has a genuine global positioning system coordinate. As it has already been stated in the “Materials and Methods” section, “Data management and statistical analysis” subsection, data was collected using the CommCare digital app that has built-in features to support the global positioning system data collection at the field level. Therefore, our data has genuine GPS coordinates.

Therefore, before we performed the spatial analyses, we did the following. We first identified Measles vaccination dropout children among the total number of eligible children included in our study. If a child received MCV1 but not MCV2, that child was considered to have dropped out of the measles vaccination program. Then, we mapped those children who were Measles vaccination dropouts.

Reviewer 2 Report

Comments and Suggestions for Authors

No further review comments for the authors. 

Author Response

Reviewer 2: No further review comments for the authors. 

Round 3

Reviewer 1 Report

Comments and Suggestions for Authors

I have one additional comment on your hotspot map (Figure 4).

Now I understand that each point on this map represents one measles vaccination dropout case and does not have a value (rate) with it. So, you were detecting the hotspots of dropout cases, not areas of high dropout rates. In the Methods, you defined the measles vaccination dropout in your study as "the percentage of children who did not receive MCV2 among those who received MCV1", and your spatial autocorrelation analysis was to examine "the distribution pattern of measles vaccination dropout across the study area". Therefore, I assume your study was designed to look at a distribution pattern of the 'percentages' but not the 'cases'. That's the reason I asked why you didn't aggregate your point data into a geographic unit by calculating the percentage of dropouts. In that scenario, your Figure 3A is supposed to be a distribution map of dropout rates at certain geographic unit (e.g., enumeration area) level, instead of a map of points (dropout cases). That will be more informative.

Author Response

Dear Reviewer,

Thank you so much for your comment/suggestion. We have revised our paper based on your comments/suggestions. Please see below the responses to the comment/suggestion you raised.

Comment:

Now I understand that each point on this map represents one measles vaccination dropout case and does not have a value (rate) with it. So, you were detecting the hotspots of dropout cases, not areas of high dropout rates. In the Methods, you defined the measles vaccination dropout in your study as "the percentage of children who did not receive MCV2 among those who received MCV1", and your spatial autocorrelation analysis was to examine "the distribution pattern of measles vaccination dropout across the study area". Therefore, I assume your study was designed to look at a distribution pattern of the 'percentages' but not the 'cases'. That's the reason I asked why you didn't aggregate your point data into a geographic unit by calculating the percentage of dropouts. In that scenario, your Figure 3A is supposed to be a distribution map of dropout rates at certain geographic unit (e.g., enumeration area) level, instead of a map of points (dropout cases). That will be more informative.

Answer:

The results section of our paper presents both measles vaccination dropout estimates and measles vaccination dropout cases.  Measles vaccination dropout estimates by target study settings have already been presented under Table 3. As indicated in the study design and settings section, we categorized our study settings into eight domains; urban slums, conflict-affected regions, IDPs, hard-to-reach areas, pastoralist populations, developing regions, newly-established regions, and refugees. Therefore, we calculated and presented the measles vaccination dropout estimates across the target settings in Ethiopia (Table 3). To avoid redundancy, we did not map vaccination dropout estimates. The way these target settings correspond to the different regions of Ethiopia is described in Table 1. The definition of measles vaccination dropout as "the percentage of children who did not receive MCV2 among those who received MCV1" mainly applies to the measles vaccination dropout estimates. In addition, we also mapped measles vaccination dropout cases by zonal administrative units (Figures 3a and 4).

We used enumeration areas for sample size calculation, which is why we did not map dropout cases by enumeration areas. Major administrative units of Ethiopia include regions, zones and districts. For our study to contribute towards improving Ethiopia’s measles vaccination coverage by providing evidence for targeted interventions, we found mapping the distribution of measles vaccination dropout cases by zonal administration meaningful and informative.  

We have revised the methods and results sections and added ‘measles vaccination dropout estimate/s’ and ‘measles vaccination dropout case/s’ (where applicable) to clearly distinguish the two terms. Changes/additions are shown in track changes.

Kind regards,

Fisseha Shiferie

Round 4

Reviewer 1 Report

Comments and Suggestions for Authors

One last comment:

Since Figure 3A is repetitive compared to Figure 4 and does not provide additional information, I would suggest changing Figure 3A to a thematic map by summarizing the number of cases in each zone and plotting them by polygons. This will help highlight zones that have a high number of dropout cases.

Author Response

Dear Reviewer,

Thank you so much for your comment/suggestion. The changes we made are shown in track changes in the revised manuscript. Please also see below a response for your comment/suggestion.

Comment:

Since Figure 3A is repetitive compared to Figure 4 and does not provide additional information, I would suggest changing Figure 3A to a thematic map by summarizing the number of cases in each zone and plotting them by polygons. This will help highlight zones that have a high number of dropout cases.

Answer:

As we believed that Figure 4 indirectly shows the distribution of measles vaccination dropout cases at zonal administrative units, we did not intend to prepare a separate map to show measles vaccination dropout cases. We included Figure 3A per your comment i.e. you requested us to first show the distribution of dropout cases at the zonal level before we performed the spatial autocorrelation test.

Zones that have a high number of measles vaccination dropout cases have already been indicated in Figure 4. As can be seen from Figure 4, the highest burden of measles vaccination dropout cases were detected in the Afar Region’s zones 1 and 5, the Amhara Region’s North Gondar Zone and peripheral areas in the Benishangul Gumuz Region’s Assosa Zone. Therefore, instead of changing Figure 3A to a thematic map, we agreed to remove Figure 3A. In other words, changing Figure 3A to a thematic map by summarizing the number of cases in each zone and plotting them by polygons will not provide additional information as zones that have a high number of dropout cases have already been shown in Figure 4.

Kind regards,

Fisseha Shiferie
